# The Importance of the Epi-Transcriptome in Translation Fidelity

**DOI:** 10.3390/ncrna7030051

**Published:** 2021-08-27

**Authors:** Charlène Valadon, Olivier Namy

**Affiliations:** Institute for Integrative Biology of the Cell (I2BC), CEA, CNRS, Université Paris-Saclay, 91198 Gif-sur-Yvette, France; charlene.valadon@i2bc.paris-saclay.fr

**Keywords:** RNA modifications, ribosomes, tRNA, translation fidelity, m^6^A, PSI, Inosine, 2′-O-methylation

## Abstract

RNA modifications play an essential role in determining RNA fate. Recent studies have revealed the effects of such modifications on all steps of RNA metabolism. These modifications range from the addition of simple groups, such as methyl groups, to the addition of highly complex structures, such as sugars. Their consequences for translation fidelity are not always well documented. Unlike the well-known m^6^A modification, they are thought to have direct effects on either the folding of the molecule or the ability of tRNAs to bind their codons. Here we describe how modifications found in tRNAs anticodon-loop, rRNA, and mRNA can affect translation fidelity, and how approaches based on direct manipulations of the level of RNA modification could potentially be used to modulate translation for the treatment of human genetic diseases.

## 1. Introduction

All types of RNA are subject to post-transcriptional modification. Since the discovery of RNA modifications in 1951, more than 150 RNA modifications have been found in coding and non-coding RNAs ranging from the addition of simple groups to the addition of highly complex structures (Figure 1) [1,2,3].

Their biological consequences are largely unknown, but the discovery that RNAs undergo dynamic, reversible chemical modifications marked the birth of the era of epi-transcriptomics. All aspects of RNA metabolism can be affected by RNA modifications [4], either directly, through changes to RNA folding or stability, or indirectly, through the action of “reader” proteins [5,6,7]. Several excellent reviews have already described in detail the role of RNA modifications in cancers and in cell-fate determination [8,9,10,11,12]. In this review, we mainly focus on the importance of RNA modifications in the two most abundant non-coding RNA families (transfer RNAs and ribosomal RNAs) and their consequences for translation fidelity. Translation proceeds via four steps: initiation, elongation, termination and recycling [13]. Generally, it begins with the fixation of the 43S ribosomal complex to the cap-binding protein eIF4F with the help of numerous translation initiation factors (eIFs) [14,15], followed by scanning of the pre-initiation complex (PIC) to the start codon [16]. The two ribosomal subunits are assembled when the PIC is present at the initiation codon by the universally conserved GTPase eIF5B [17,18]. Elongation begins with the delivery of a tRNA to the ribosomal A-site by elongation factors [19] and continues codon-by-codon until the ribosome reaches a stop codon. When a stop codon enters the ribosome, it is recognized by the termination complex eRF1/eRF3, causing the release of the peptide [20,21]. RLI1/ABCE1 splits the ribosome into its two subunits [22,23], which are then available for a new translation cycle [24].

Translation is not perfectly accurate, as it has a median error rate of 0.01% in humans [25,26]. There are 30 codons in the human code that depend on the incorporation of a near-cognate tRNA (pairing of two of the three bases). Codon-anticodon pairing is known to be flexible at the third position of the codon, but it is clear that RNA modifications alter translation accuracy [27]. The incorporation of a near-cognate tRNA can occur during elongation or termination, in which case it is known as readthrough [28].

In this review we will describe modifications affecting two abundant types of non-coding RNAs—tRNAs and rRNAs—and will consider how such modifications fine-tune translational accuracy. We will also discuss the importance of certain mRNA modifications affecting ribosome fidelity. There is a striking difference in the mode of action of modifications between coding and non-coding RNAs, in that modifications to rRNAs and tRNAs act directly on the folding and activity of the molecule, whereas most of the modifications to mRNAs act via reader proteins. This simplified presentation needs to be modulated because some tRNA modifications are required for the proper action of aminoacyl-tRNA synthetases (aaRS) [29].

## 2. Control of Translation Fidelity by Modifications to Cytosolic tRNAs

The type of RNA most frequently modified in cells is tRNA, in which about 17% of nucleotides are modified [30]. Modifications have been found in all five domains of tRNA (i.e., acceptor stem, D-loop, T-loop, V-loop and anticodon loop) (Figure 2).

Exceptions exist, but most of the modifications to the D- and T-loops affect the stability or folding of the molecule, whereas those in the anticodon region can affect either the recognition by aaRS or the fidelity of genetic code translation. The anticodon region occupies positions 34-35-36 (Figure 2) that directly base pair to the mRNA codon but are also recognized (together with the acceptor stem) by some aaRS [29]. Some modifications found in the anticodon loop specifically alter the reading frame maintenance. This is the case at position 37, which has been reported to affect the maintenance of the reading frame with the wybutosine yW modification [31]. Interestingly, in humans, no modification has yet been found at position 36, which pairs with the first position in the codon, whereas position 34 of tRNAs, which pairs with the third nucleotide of each codon, is one of the positions at which the various chemical modifications are most numerous [3,32,33]. The reasons for these differences remain unclear, but may reflect the importance of strict base-pairing at certain positions to prevent incorrect amino-acid incorporation, whereas flexibility may be more acceptable at other positions, at which it may not necessarily cause a change of amino acid, thanks to the redundancy of the genetic code. For a long time tRNA modifications have been considered irreversible. However, in 2016, the work of Fange et al. demonstrated that *ALKBH1* can remove the methyl group from m^1^A_58_ in tRNAs [34], opening the possibility that tRNA modifications would be more dynamic than anticipated. Below, we review modifications of the anticodon loop known to affect the translation of the genetic code by affecting the efficacity of cognate or near-cognate tRNA incorporation.

### 2.1. mcm^5^U_34_ Modifications

The modifications observed at the U_34_ position of tRNA are 5-methoxycarbonylmethyluridine (mcm^5^U), 5-methoxycarbonylmethyl-2-thiouridine (mcm^5^s^2^U) and their derivatives. Catalysis begins with the addition of 5-methoxycarbonylmethyl (cm^5^) to uridine by the elongator complex [35,36]. The cm^5^U is then further modified by the addition of a methyl group by a heterodimeric complex, Trm9-Trm112 (*ALKBH8* in mammals) [37]. Finally, the oxygen atom attached to the C2 atom of the uracil ring may undergo thiolation in a subset of tRNAs, catalyzed by the Ncs2-Ncs6 complex and resulting in a final mcm^5^s^2^U modification [38]. Ultimately, the U_34_ position is modified in most eukaryotic tRNAs.

The roles of mcm^5^U and mcm^5^s^2^U in codon–anticodon recognition have been studied in depth in yeast [39,40,41]. The rate of amino-acid misincorporation has been assessed by dual-luciferase reporter assays in a *S. cerevisiae TRM9* mutant (absence of mcm^5^U and mcm^5^s^2^U) [40]. The Trm9 deletion decreases the fidelity of translation specifically for the Arg, Gln, Glu and Lys tRNAs. U_34_ modifications, thus promoting discrimination between some cognate and near-cognate codons. A second study in *S. cerevisiae* used mass spectrometry to specify amino-acid misincorporation during readthrough, in the presence or absence of U_34_ modifications. A similar phenotype was observed for tRNA_Arg_ under *TRM9* gene deletion. However, the results obtained clearly differed from those for tRNA_Gln_, for which U_34_ modifications of Gln enhance misincorporation. The impact of ALKBH8 protein deficiency has been investigated in mammals [37], through the generation of *Alkbh8^−/−^* mice. This mouse line has no mcm^5^U, mcm^5^s^2^U, or mcm^5^Um modifications to tRNAs, these modifications being replaced by the corresponding acid/amide forms: cm^5^U and/or ncm^5^U/ncm^5^s^2^U. Hypomodification of the selenocysteine tRNA (tRNA_Sec_) impairs its ability to decode the UGA stop codon in vitro. U_34_ modifications therefore play a role in codon–anticodon recognition in mammals, as in the yeast model. Despite the availability of this mammalian model, studies of the impact of U_34_ modifications on other tRNAs have yet to be published.

The physiological importance of U_34_ modifications has been demonstrated for mcm^5^s^2^U. This is especially well illustrated by the finding that loss of U_34_ modifications can lead to ribosome pausing, promotes proteotoxic stress and protein aggregation [42]. Indeed, the presence of the sulfur atom provides an extended chemical group, which stabilizes A-U or G-U pairing [43]. In conclusion, the weak interaction between A-U is strengthened by U34 modification, favoring translation fidelity, whereas stabilization of the unconventional G-U base-pairing favors the incorporation of near-cognate tRNAs.

### 2.2. I_34/37_

Inosine (I) results from deamination of the C6 of adenine [44]. Its editing is catalyzed by adenosine deaminases, which act directly on tRNA (ADATs) [45]. Inosine is present at tRNA positions 34 (8 tRNAs) and 37 (tRNA_Ala_) in eukaryotes [45,46,47]. I_34_ is catalyzed by the heterodimeric enzyme ADAT (hetADAT), consisting of ADAT2 and ADAT3. I_37_ is deaminated by *ADAT1* and further modified by methylation (m^1^I_37_) catalyzed by the tRNA methyltransferase *TRM3* [48]. A deficiency of I_34_ has been reported to affect human health, with patients presenting bi-allelic *ADAT3* mutations displaying intellectual disability [49].

I_34_ has been shown to enable tRNAs to pair with U, A and C nucleotides at the wobble position [27]. The chemical origin of the I_34_ base-pairing effect is the replacement of the hydrogen-donating amino group in the C6 position by a hydrogen-accepting oxygen [44]. Inosine contributes to the extension of the genetic code [50].

Presence of inosine in humans has been linked to a deviation of codon usage between prokaryotes and eukaryotes [51]. Indeed, bioinformatic analysis has shown that TAPSLIVR amino-acid stretches are more efficiently expressed with I_34_-tRNAs. This has resulted in a greater abundance (eight-fold) and length of such proteins in eukaryotic proteomes than in prokaryotic proteomes, which have only one I_34_-tRNA.

### 2.3. Q_34_ and Its Derivatives

Queuosine (Q), or 7-deazaguanosine, is a modified analog of guanosine incorporated at G_34_ of GUN anticodon tRNAs [52]. In mammals, Q_34_ is hypermodified at the Tyr and Asn anticodons, by the addition of a sugar (a galactose (GalQ) and a mannose (ManQ), respectively) to the C4 hydroxy group of the cyclo-pentenediol (Figure 1) [53,54,55]. Eukaryotes cannot synthesize Q de novo, and must therefore obtain it (or its derivative, queuine) as a micronutrient from the gut microbiota or through dietary intake [56,57,58]. Human cells must, therefore, take up queuine, to replace guanosine by queuosine. Interestingly, queuosine levels change during development [59,60]. It remains unclear whether these changes reflect differences in the need for translation fidelity during development or whether they simply reflect the availability of queuine in the diet and/or the possibility of its synthesis by the microbiota. Queuosine has also been associated with cell differentiation, the symptoms of poly-phenyl-ketonurea disease, cancer progression and microbiote diversity. Mannosyltransferase responsible for the generation of ManQ has been isolated from rat liver, but no galactosyltransferase has been identified yet for GalQ [61]. It remains unclear whether these sugar-modified tRNAs are involved in translation (i.e., are they still able to enter the ribosome?) or whether they act as regulatory RNAs, as reported for tRNA-derived fragments [62].

Despite its key position in the anticodon, the role of Q in translation fidelity has been little studied. The impact of Q on codon–anticodon pairing has been physically measured; the C-G pairing is slightly more stable than Q-G, and the Q-U pairing is more than twice as stable as C-U [63]. However, it remains unclear how Q is distinguished from G in front of C or U [64]. Q has been shown to modulate translation accuracy in *E. coli* [65]. However, a study in eukaryotes investigated the role of Q in tRNA_His_ decoding in *Xenopus laevis* oocytes [66]. GUG or QUG—tRNA_His_ from *D. melanogaster* was injected into the oocytes, and their ability to decode CAC or CAU codons was assessed. The results underline that Q_34_-tRNA_His_ decoded CAU more efficiently than the CAC codon, contrary to what was found for G_34_-tRNA_His_. Additional studies are required to clarify the role of Q and to characterize the role of its hypermodified derivatives, GalQ and ManQ, in translation fidelity.

### 2.4. m^5^C_34_

5-methylcytosine (m^5^C) is an additional methyl group on C5 present on the cytosine at the position 34 of tRNA_Leu_(CAA). In humans, the methyltransferase hTrm4 is responsible for this methylation position [67], whereas its yeast homolog Trm4 also modifies positions 48 and 49 [68]. In *S.cereviae*, tRNA_Leu_(CAA) is surnamed “tRNASUP53” for its abilities to suppress the UAG stop codon. Interestingly, the presence of *m^5^C_34_* on tRNASUP53 depends on the integrity of a 32 bases intron in the pre-tRNA.

The importance of *m^5^C_34_* in tRNA_Leu_(CAA) mis-incorporation has been assessed in the yeast model [68]. The suppressor activity of tRNASUP53 has been tested on medium minus tryptophan, using a trpl-J(Am) marker. The results showed that in the absence of *m^5^C_34_* the cells do not grow on the medium, highlighting a decrease in the tRNASUP53 suppressor activity. Thus, *m^5^C_34_* in tRNA_Leu_(CAA) is considered to act as an enhancer of tRNA misincorporation modification. Unfortunately, for now, no replica of these results exists in a human or another eucaryotic model. 

The lack of interest in this modification could be due to the low level of leucine incorporation at the UAG stop codon which has not been reported in any recent study; moreover, hTrm4 has not been linked to any disease in humans.

### 2.5. ms^2^t^6^A_37_

2-methylthio-N6-threonylcarbamoyladenosine (ms^2^t^6^A) biosynthesis is well known: the N6-threonyl carbamoyl adenosine (t6A) is methyl-thiolated to generate the ms^2^t^6^A at position 37 of the tRNA_Lys_(UUU) which is the only tRNA concerned by this modification in humans. The methyl-thiolation is done by *Cdkal1* [69]. There is some structural evidence that ms^2^t^6^A at position 37 inhibits a noncanonical U33-A37 interaction and is likely to be needed to compensate for the relatively weak U-turn remodeling properties of mcm5s2U_34_ [70].

For now, studies of the role of ms^2^t^6^A in translation fidelity have been performed only in a bacteria model [71]. In *B. subtilis,* absence of mcm5s2U_34_ (ΔyqeV) revealed that ms^2^t^6^A_37_ in tRNA_Lys_(UUU) prevents the misreading of its cognate codons AAA and AAG, especially when the rate of translation is high. Unfortunately, ten years after this study, these encouraging data have not yet been reproduced in a eukaryotic model. However, in the same study, a knock-out of *Cdkal1* in mouse has been performed and resulted in reduction of glucose-stimulated proinsulin synthesis. Thus, it has been hypothesized that, in the absence of *Cdkal1*, tRNA_Lys_(UUU) misreading increases, limiting the generation of mature insulin and C-peptide which relies on Lys 1 and 2 of proinsulin. This phenotype could explain the molecular pathogenesis of type 2 diabetes in patients carrying *Cdkal1* risk alleles, new evidence for the importance of tRNA modifications in human health.

### 2.6. i^6^A_37_

N6-isopentenyladenosine (i^6^A_37_) results in the addition of an isopentenyl group onto N^6^ of A_37_ by an isopentenyl-transferase (IPTase). The nature of concerned tRNAs varies from one organism to another. In humans, i^6^A_37_ has been found to be synthetized by *TRIT1* in cytosolic tRNA_Ser_ (UGA, AGA, and CGA) and tRNA_[Ser]Sec_ [72]. The last is relatively hypomodified (~40%). As a member of near-wobble modification, I^6^A_37_ is proposed to enhance A:U stacking by stabilizing the Watson-Crick base pair but the chemical reasons are not yet clear [73].

The role of i^6^A_37_ on translation fidelity was assessed in a *S. cerevisiae* model (Blanchet 2017). Mod5p enzyme synthetizes the i^6^A_37_ in the Tyr and Cys tRNAs. The ability of Tyr and Cys tRNAs to readthrough respectively UAA/UAG and UGA stop codons has been assessed in a ΔMod5p strain by mass spectrometry. In this strain, both tRNAs are less efficient at being mis-incorporated, revealing that i^6^A_37_ modification acts as an enhancer of translation plasticity in *S. cerevisiae*. In another study in *S. pombe*, the role of i^6^A_37_ has been assessed using a β-galactosidase codon-swap reporter [74]. This confirms that i^6^A_37_ increases the incorporation of tRNA_Tyr_ at a near-cognate codon. Moreover, it shows that the modification enhances the incorporation of tRNA_Cys_ at a cognate codon. Altogether, these data suggest that i^6^A_37_ promotes decoding activity generally. 

To date, no study has been made in humans about the role of i^6^A_37_ in translation fidelity. However, it is clear that *TRIT1* mutation is associated with severe diseases. Indeed, it has been defined as a tumor suppressor [75]. The mutation of this protein is also associated with encephalopathy and myoclonic epilepsy pathology [76]. 

### 2.7. Ψ35/38/39

Pseudouridine (Ψ) is an isomer of uridine in which uracil binds the ribose via a C1′-C5 rather than a C1′-N1 bond [77]. The pseudouridine synthases responsible for the catalysis of tRNA Ψs belong to the PUS RNA-independent family. Ψs are mainly found in the anticodon loop of tRNAs, at positions 35, 38 and 39. Catalysis is performed by two enzymes: PUS7 and PUS3. The effects of Ψ_35,_ Ψ_38_ and Ψ_39_ on translation fidelity probably stems from their ability to improve the stacking of double-stranded RNA over that achieved with the uridine isomer [78]. More precisely, the presence of Ψ stabilizes the C3′-endo conformation, creating an additional N1 H-bond donor [79]. Thanks to these chemical characteristics, Ψs involved in (Ψ_35_) or next to (Ψ_38_-Ψ_39_) the anticodon enhances the stability and structure of pairing. 

Studies in yeast (*S. cerevisiae*) have shown that the deletion of *PUS3* triggers an increase in misincorporation events relative to the WT strain [39,80]. Even Ψ_38_ and Ψ_39_, at the end of the anticodon loop increase miscoding frequency. We have also studied the contribution of Ψ_35_ to the ability of tRNA_Tyr_ to act as a near-cognate tRNA for stop-codon readthrough in a Δ*PUS7* strain [39]. Mass spectrometry has shown that tyrosine incorporation is less efficient in the absence of the *PUS7* gene. This finding highlights the importance of Ψ_35_ for the ability of tRNA_Tyr_ to read through UAA and UAG codons. Together, these studies demonstrate that Ψ modifications to the anticodon loop of tRNAs have a major effect on translation fidelity. The tRNA pseudo-uridine synthases *PUS3* and *PUS7* have both been implicated in human health problems; they cause different diseases, and notably intellectual disability [81,82].

It is interesting to mention that some modifications away from the anticodon loop seems to play a role in translation. For example, absence of ac4C_12_ and dU_20_ of Leucine et Serine tRNAs causes a reduction in A-site occupancy at the corresponding codons. Widespread changes in the A-site occupancy have been also observed in the absence of m^2^_2_G_26_ [83]. Since these modifications are not in the anticodon loop, they may influence either the charging of these tRNAs or their ability to bind stably the ribosomal A-site. Several studies suggest that interactions inside the anticodon loop are crucial for a correct modification of the tRNAs. These interconnections between modifications create a complex network. For example, in eukaryotes, the yW formation at m^1^G_37_ of tRNA_Phe_ is greatly stimulated by the presence of Cm_32_ and Gm_34_ [84,85,86]; the presence of i^6^A_37_ or t^6^A_37_ in tRNA_Ser_ stimulates the formation of m^3^C_32_ [87]; m^5^C_38_ is stimulated by prior Q_34_ formation [88]; and I_34_ editing in tRNA_Thr_(AGU), is stimulated by prior C to U editing at position 32 [89]. Future structural studies of tRNA-modification enzymes complexes will probably explain such dependencies.

It is also possible that tRNA modifications act through other processes such as RNA sequestration, or the generation of tRNA-derived fragments (tRFs/tiRFs) [90] obtained by the endonuclease cleavage of tRNAs (mainly in the D-Loop, TΨC-Loop and anticodon-Loop), either in normal or stressed conditions. Such tRFs/tiRFs are involved in various physiological and pathological processes by modulating RNA stability or translation [62]. There is a clear link between tRNA modifications and tRFs/tiRFs generation. For example, Dnmt2, which methylates the C_38_ of tRNA_Asp_ [91]_,_ protects tRNA from the degradation [92]. It has been also shown that *ALKBH3* activity results in removal of m^1^A and m^3^C modifications, leading to a sensitivity to angiogenin cleavage [93].

## 3. Role of rRNA Modifications in Translation Fidelity 

Ribosomal RNA is the most abundant non-coding RNA in the cytoplasm. It is the main constituent of the ribosome. In total, 200 modification sites have been mapped on the human ribosome, in which about 2% of the nucleotides are modified [30,94,95,96]. These modifications can modulate all stages in the life of the rRNA, from ribosome biogenesis to translation accuracy [97]. The most frequent modifications observed are pseudo-uridines and 2′-O-methylations, although base methylation and acetylation have been reported [30]. We focus here on the description of the two main modifications of rRNAs known to affect translation fidelity.

### 3.1. 2′-O-methylation (Nm)

2′-O-methylation (Nm) is a modification in which a sugar is added to the 2′C hydroxyl group of the nucleotide. The chemical impact of Nm on RNA has been investigated by several studies. It has been reported that Nm biases the sugar pucker equilibrium in favor of the C3′-endo conformation of pyrimidines [98]. Intra-residue steric repulsion occurs between the Nm, the 3′-phosphate, and the 2-carbonyl groups in the C2′-endo conformation, favoring the C3′ form. The Nm modification may, therefore, either stabilize or modulate RNA structures.

In human cells, Nm is mediated by the ribonucleoprotein complex consisting of the methylase fibrillarin (FBL) and the guide RNA (C/D box snoRNA) specific to the methylation site [99]. FBL is an essential protein, but it can be partially inactivated, leading to a decrease of up to 50% in the number of methylation sites in human cells [100,101]. More than 100 2′-O-methylation sites have been mapped on rRNAs, independently of nucleoside identity [102,103].

The role of Nm in miscoding has been explored in human cancer cells [104]. FBL overexpression, leading to hypermethylation of the ribosome, has been shown to trigger an increase in amino-acid incorporation at cognate or near-cognate codons. It is difficult to identify the 2′-O-methylation sites responsible for this phenotype, because site-specific inactivation experiments have not been performed yet on human cells. As FBL methylates all the sites, the only solution would be to inactivate each snoRNA specifically, one-by-one. A study of this type has been performed in yeast, in which knockouts of the various guide C/D box snoRNAs have been performed [105]. The impact of the loss of each snoRNA was evaluated by measuring stop codon readthrough efficiency. Nm-C_1639_ was identified as the most important of the Nm sites tested. The abolition of Nm at this P-site triggers a slight increase in UAG readthrough. This work revealed a role for Nm-C_1639_ in the maintenance of ribosome fidelity during termination. There is now a need to reproduce such systematic analyses of Nm sites in humans.

The role of rRNA’s Nm extends beyond miscoding events. The downregulation of FBL has been shown to alter IRES-dependent initiation and frameshifting. A single deletion of Am_398_ or Gm_3745_ in the 28S rRNA or of Am_163_ in the 18S rRNA is embryo-lethal in zebrafish [106]. Moreover, FBL overexpression has been reported during the differentiation of human stem cells, and in several cancer studies, suggesting a central role in these processes [100,104,107,108].

### 3.2. Pseudouridine 

With the exception of position 50 in the 5S rRNA that is catalyzed by the enzyme *PUS7*, the formation of Ψs in rRNA is catalyzed by a ribonucleoprotein complex composed of the pseudo-uridine synthase DKC1 associated with H/ACA box snoRNAs [109,110]. In human rRNAs, Ψs are mapped with a Ψ/U ratio of 5–7%, with a total of about 100 sites [109,111,112,113,114]. DKC1 is as an essential protein, and mutations of its gene have been linked to X-linked dyskeratosis congenita disease. Patients may display alterations to skin color, nail dystrophy, bone marrow failure, and an increase in the risk of developing cancer and pulmonary fibrosis, although it is not clear whether these effects are related to the absence of Ψ from rRNA [110].

The role of Ψs in miscoding has been investigated in human cells [115]. SNORA24 (ACA24), a H/ACA box snoRNA guiding the Ψ_609_ and Ψ_863_ on the 18S rRNA, has been downregulated in HCC cells [116]. An analysis of ribosomal pre-translocation complex dynamics by sm-FRET indicated changes in tRNA conformation in the A-site in ribosomes lacking Ψ_609_ and Ψ_863_ relative to wild-type ribosomes, depending on the tRNA entering the ribosome. It has also been shown that lower levels of SNORA24 expression increase amino-acid misincorporation by 10%–20% and readthrough by 15% at UGA, but not at UAG codons.

The way in which Ψs in rRNAs decrease the accuracy of translation seems to depend on their abundance in the peptidyl transferase and decoding centers of the ribosome [77]. Ψs are known to generate an additional N1 H-bond donor and to stabilize the C3′-endo conformation [79]. This enables Ψs to increase RNA–RNA stability in the fidelity centers of the ribosome [117]. A decrease in the number of Ψ sites is, thus, accompanied by ribosome destabilization, resulting in a decrease in ribosome fidelity.

## 4. mRNA Modifications Influence the Reading of the Genetic Code

Many studies over the last decade have revealed the importance of mRNA modifications. These modifications are highly dynamic, with eraser proteins able to eradicate the modifications from the mRNA. The dynamic aspect of the modifications allows integration in a very efficient manner of the RNA metabolism and translation to the physiological state of the cell, considering the appearance of possible stresses.

### 4.1. Inosine

The formation of inosine on mRNAs is catalyzed by the adenosine deaminases ADAR1 and 2 [118]. The inosines of mRNAs, like those of tRNAs, play a major role in expansion of the genetic code, with 5072 identified editing sites in human coding sequences [119]. 

One of the best known examples of the importance of A-to-I editing in mRNA is the modification of the glutamate receptor subunit B (GluRB) precursor messenger RNA: CAG (Gln) → CIG (Arg) in exon 11. This site is modified by ADAR2 and is essential to ensure the impermeability of the glutamate receptor to Ca^2+^ ions [120]. A defect of this inosine site has, notably, been shown to contribute to neuronal death in amyotrophic lateral sclerosis [121]. The dysregulation of ADAR1 and 2 has also recently been observed in human hepatocellular carcinoma [122]. Patients with an upregulation of ADAR1 and a downregulation of ADAR2 have higher incidences of tumor recurrence and liver cirrhosis, and shorter disease-free survival times. These dysregulations are linked to changes in the number of inosine sites, with, in particular, hyper-editing of the *FLNB* mRNA and hypo-editing of the *COPA* mRNA [122]. Finally, ADAR1 seems to act as an oncogene, whereas ADAR2 acts as a tumor suppressor, in hepatocellular carcinoma.

Inosine in mRNAs is known to modulate alternative splicing and stability, but it clearly also plays an essential role as an enhancer of near-cognate tRNA incorporation, ensuring the activity of some proteins [118]. On the other hand, we did not find any significative difference in ribosome profiling between edited and non-edited mRNA in term of translation efficiency in *A. Thaliana* mitochondria [123]. The conservation of these essential CDS sites, rather than the cognate codon with a G, remains to be evaluated.

### 4.2. Pseudouridine

Unlike the Ψs found in rRNA, the reaction generating those found in mRNA is catalyzed by pseudo-uridine synthases, which are RNA-independent proteins [113,124,125,126], although the existence of some box H/ACA snoRNAs complementary to mRNAs raises the possibility that RNA-dependent pseudo-urylation of mRNAs also occurs [127,128]. mRNA Ψs are known to be modulated under cellular stress and during development, but no Ψ reader or eraser has yet been described [129]. Within the translated and untranslated regions of mRNAs, pseudo-uridine is present with a Ψ/U ratio of 0.2–0.6%, and 1889 sites have been identified by N3-CMC–enriched pseudo-uridine sequencing [113]. More than 60% of pseudo-uridine residues are located within the coding sequence, suggesting a link with translation [130,131].

In prokaryotes, several studies have described the ability of Ψ to alter base-pairing and induce misincorporation [132,133,134]. However, far fewer studies have been performed on human cells [131]. Amino-acid misincorporation in front of a “U-codon” has been shown to occur at a rate of 1%. The presence of Ψ in mRNA induces the substitution of Ser, Ile or Leu for Phe at UUU/C codons; Cys or His substitution for Tyr at UAU/C codons; and Pro or Gln substitution for Leu at CUA/U/C/G codons (Figure 3).

Given the high frequency of Ψ in mRNA and its role in near-cognate tRNA recognition, Ψ modifications probably make a major contribution to translation fidelity. A closer look at codon/anticodon base-pairing in the case of the misincorporation of Cys at a Tyr codon reveals a central mismatch between an A and a C. This unfavorable interaction is probably compensated for by the strong ability of Ψ to stabilize the codon/anticodon structure by stacking interactions. Indeed, Ψ is known to enhance RNA structure stability. Despite its ability to form a supplementary N1-hydrogen bond, Ψ has the same Watson-Crick base-pairing properties as U [79]. 

### 4.3. m^6^A

The N6-methyladenosine (m^6^A) modification involves the addition of a methyl group to the N atom linked to the C6 of adenosine. Chemical predictions of the impact of m^6^A on RNA–RNA base-pairing suggest a disruption of this interaction due to the methyl group [135]. Indeed, this group must adopt an anti-conformation in the context of A–U pairing. This conformation is less energetically favorable than the syn conformation, leading to destabilization of the RNA–RNA accommodation. 

In humans, more than 12000 m^6^A sites are estimated to be present on 7000 mRNAs [136,137]. About 35% of m^6^A sites are located within the coding region [138]. m^6^A is a dynamic modification that has been reported to interact with several enzymes called readers [139,140]. A heterodimeric methylase complex (*METTL3*-*METTL14*) is responsible for adding the methyl group. Once modified, the site can be recognized by reader proteins, most of which belong to the YTH-domain protein family (YTHDC and YTHDF), or eraser proteins, which are demethylases (such as *FTO* and *ALKBH5*). 

The impact of m^6^A at the first or second position of the codon has been measured by quench flow techniques [141]. This modification delays tRNA incorporation, by slowing tRNA accommodation at site A of the ribosome. However, it has been reported that m^6^A at the middle position of the codon has a lesser effect on pairing for near-cognate than for cognate tRNAs [142]. This difference in kinetics suggests that tRNA misincorporation rates are likely to be higher in the presence of m^6^A at the middle position. However, contrary to these findings for prokaryotic systems, mass spectrometry assays in eukaryotes (wheat germ and HEK293T) identified no miscoding effect of the m^6^A modification [143,144]. The method used for eukaryote systems may be insufficiently sensitive to detect misincorporation in the context of cognate/near-cognate competition. Indeed, the same study found no miscoding effect of Ψ modification, contradicting the findings of another team published in the same year [131]. In the face of these conflicting data, further studies are required to clarify the impact of m^6^A on miscoding events.

m^6^A is one of the most commonly studied RNA modifications because of its broad influence on RNA maturation and degradation, RNA-protein interactions and translation efficiency, implicating this modification in a number of different biological processes [138,145,146,147,148,149]. Focusing on human health, altered m^6^A levels have been implicated in the regulation of the expression of genes relating to cancer pathogenesis and development [150].

### 4.4. m^5^C

5-methylcytosine (m^5^C) is a cytosine with an additional methyl group on C5. Like m^6^A, m^5^C is a dynamic modification, with writer, reader and eraser proteins. *NSUN2* and the Aly/REF export factor are the principal m^5^C mRNA writer and reader proteins, respectively [151]. m^5^C has been mapped on several transcriptomes in humans [152,153,154,155]. Although *NSUN2* and *NSUN6* are well-known tRNA-modification enzymes, they also appear to modify mRNA. The number of m^5^C sites in mRNA has been estimated at about a thousand by bisulfite RNA sequencing [156]. Interestingly, viral RNAs are particularly rich in m^5^C modifications, suggesting that it could play a role in the discrimination of endogenous and exogenous RNAs.

The question of the impact of m^5^C on translation has been addressed by ribosome profiling in Hues9 human embryonic stem cells with a knockout of *NSUN6* gene [157]. No global translational defect was observed, but the absence of *NSUN6* was found to trigger stop codon enrichment at the P-site of the ribosome, possibly after readthrough, and an increase in ribosomes bound to the 3′UTR of mRNAs modified by *NSUN6*. These data suggest that m^5^C sites in the 3′UTR of mRNA enhance translation termination efficiency by decreasing the readthrough rate. It remains unclear how m^5^C in the 3′UTR affects termination. Another study in HEK293T cells assessed the impact of m^5^C at the three codon positions by mass spectrometry [144]. None of the three positions was found to modulate the misincorporation of amino acids. 

m^5^C is linked to human health. Indeed, *NSUN2* mutations are associated with growth retardation, neurodevelopmental defects, and have been identified as a possible treatment target for tumors [153,158,159,160,161]. Moreover, the m^5^C reader and eraser proteins cited above are known to display altered expression levels in various types of cancer [151].

## 5. Manipulation of RNA Modifications to Treat Human Diseases

The field of RNA modifications is undoubtedly a very promising area in human therapy. Synthetic modified mRNAs can be used in diverse therapeutic contexts, including cardiac regeneration, asthma, cystic fibrosis or lung diseases [162,163,164,165]. The best-known application is probably the current COVID-19 vaccines of Pfizer/BioNtech and Moderna. In these mRNA-based vaccines, all the uridine residues are replaced by N1-methyl-pseudouridines to prevent the recognition of the vaccine mRNA by host RNA sensors and to stimulate translation initiation by attenuating eIF2α phosphorylation [166,167]. For those interested in eiF2α stress response and translational regulations, please see the following review [168]. It is also possible to target mRNAs directly, through the use of artificial snoRNAs to replace a U residue with a Ψ at a specific position [132]. In this example, changing the U to a Ψ at the first position of a premature termination codon leads to the incorporation of several amino acids rather than a stopping of translation. This could restore production of the full-length protein, thereby correcting the genetic defect. 

From another standpoint, RNA modifications affect diverse biological processes, and the correct incorporation of many of these modifications, at the correct sites, is required for normal development. Alterations to these modifications have been implicated in several diseases, including cancers and resistance to therapy of melanoma cells [169]. The role of m^6^A in cancer is very well documented, and m^5^C has also emerged as a major player in cancer development [170,171,172]. Given the crucial roles of writer, reader and eraser proteins in cell homeostasis, these proteins have naturally emerged as potential treatment targets [173]. Ribosome modifications are also of potential interest in this context, and DKC1 and FBL may serve as potential anticancer targets, as shown by the changes in their levels of expression in many cancers [104,174]. 

As discussed above, it is possible to target mRNA with an H/ACA snoRNA for the incorporation of a Ψ at a specific position. This approach could be used in genetic diseases caused by the presence of a premature termination codon (PTC). Proof-of-concept has been obtained through the demonstration that replacing the U of the stop codon with Ψ converts the stop codon into a sense codon [132]. Indeed, serine and threonine were found at ΨAA and ΨAG codons, whereas tyrosine and phenylalanine were found at ΨGA codons. In principle, it should be possible to change the modification status of tRNAs to modulate translation fidelity. This would be particularly useful in diseases linked to the appearance of a premature stop codon, which are treated with readthrough-inducing molecules. These molecules, such as aminoglycosides, target the ribosome, enabling it to read through the stop codon, but it should be possible to improve the incorporation of specific tRNAs by altering their modifications [175]. However, in this case, a delicate balance must be found between promoting high levels of readthrough without compromising normal tRNA usage. The recent publication describing the stimulation of UGA readthrough by inhibiting the Cm_34_ modification on tRNA_Trp_ with 2,6-diaminopurine (DAP) paves the way for the development of such therapeutic approaches [176]. We are still at the dawning of the epi-transcriptomic era, particularly as concerns human treatments, but this field promises to yield extraordinary advances. 

## 6. Conclusions

With so many unanswered questions both in terms of molecular mechanisms and physiological consequences, the field of RNA modifications gains more and more interest. One of the current limitations is the difficulty to identify and quantify RNA modifications, especially in highly structured molecules such as tRNAs or rRNAs. Mass spectrometry approaches are extremely accurate, but require highly purified molecules and are hardly quantitative on a large scale. Deep-sequencing (NGS) approaches require either chemical modification of the RNA or immunoprecipitation with a specific antibody, with the associated problems of specificity [177]. Direct RNA sequencing (nanopore) holds a lot of promise with the possibility of directly detecting modified positions. However, this still requires the development of bioinformatics tools to allow a reliable and quantitative analysis. There is no guarantee that all tRNAs will be accessible, although preliminary report exists suggesting that some will be [178].

## Figures and Tables

**Figure 1 ncrna-07-00051-f001:**
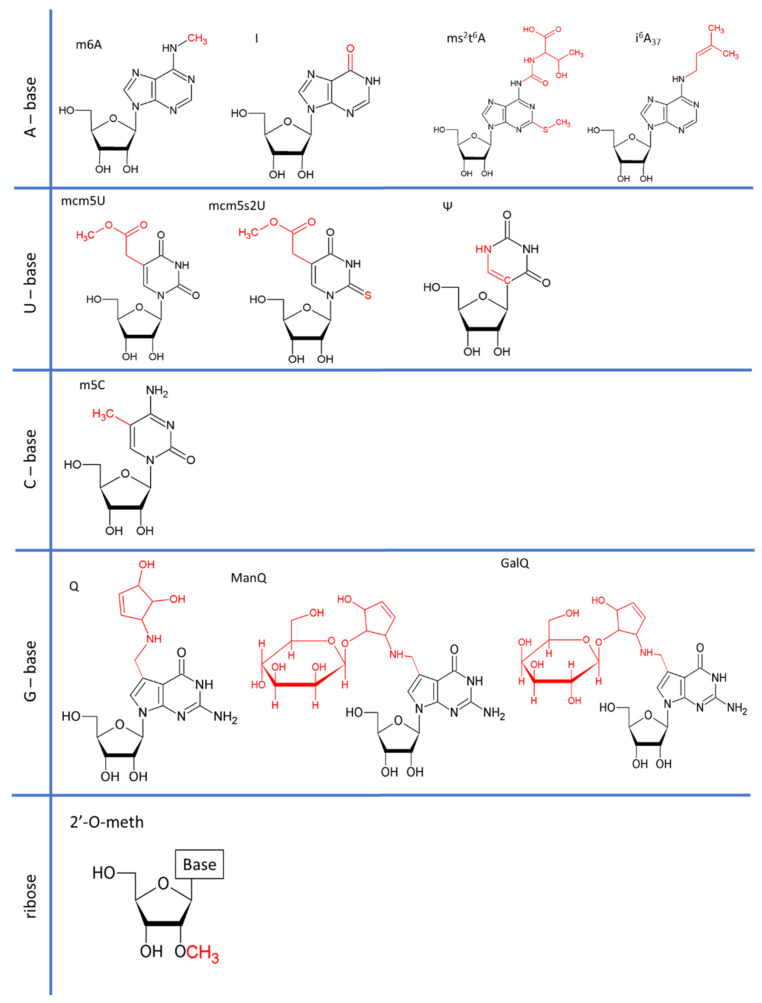
Structure of different RNA modifications discussed in this review. In black: basic structure of the base or ribose concerned. Red: chemical modification. m6A: N6-methyladenosine. I: inosine. ms^2^t^6^A: 2-methylthio-N6-threonylcarbamoyladenosine. I^6^A: N6-isopentenyladenosine. mcm^5^U: 5-methoxycarbonylmethyluridine. mcm^5^s^2^U: 5-methoxycarbonylmethyl-2-thiouridine. Ψ: pseudo-uridine. m5C: 5-methylcytosine. Q: queuosine. ManQ: mannosyl-queuosine. GalQ: galactosyl-queuosine. Nm: 2′-O-methylation.

**Figure 2 ncrna-07-00051-f002:**
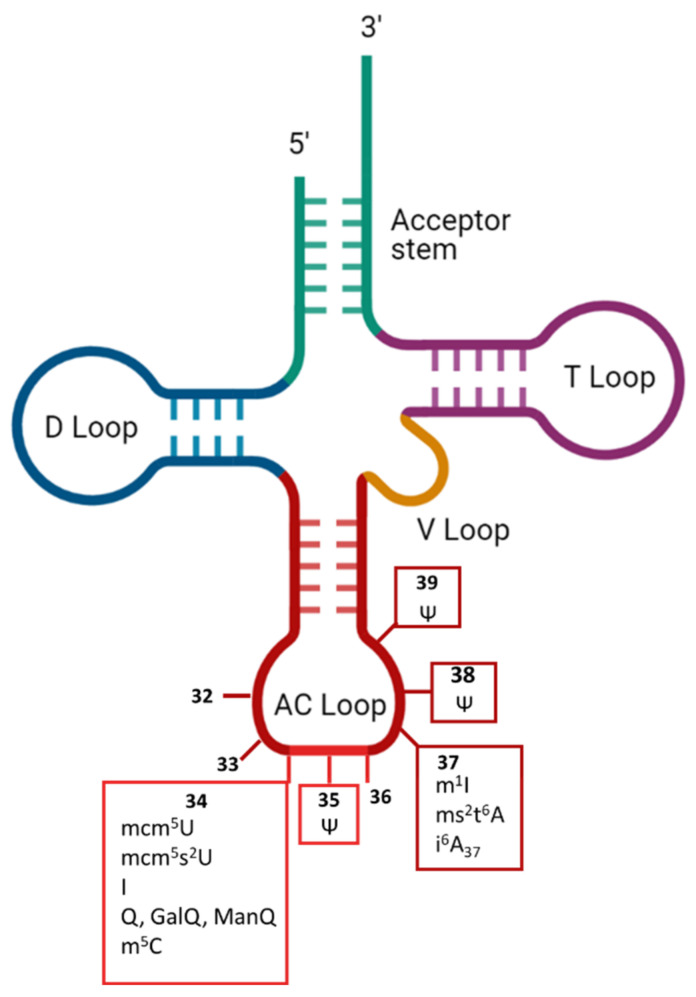
Overview of human cytosolic tRNAS anticodon loop modifications known to shape translation fidelity. AC loop: anticodon loop. Ψ: pseudo-uridine. I: inosine. m^1^I: N^1^-methylinosine. mcm^5^U: 5-methoxycarbonylmethyluridine. mcm^5^s^2^U: 5-methoxycarbonylmethyl-2-thiouridine. Q: queuosine. GalQ: galactosyl-queuosine. ManQ: mannosyl-queuosine. ms^2^t^6^A: 2-methylthio-N6-threonylcarbamoyladenosine. m^5^C: 5-methylcytosine. i^6^A: N6-isopentenyladenosine. 32–39: anticodon loop positions.

**Figure 3 ncrna-07-00051-f003:**
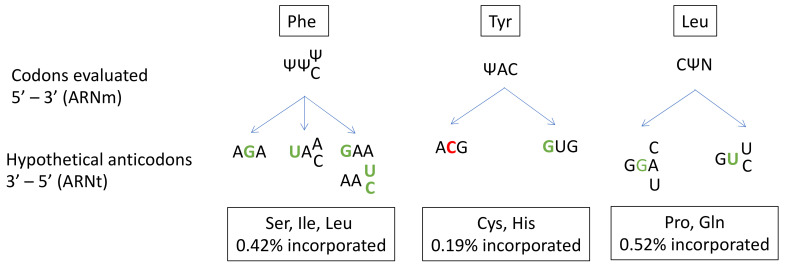
Prediction of anticodon substitution in front of codons with pseudo-uridine (Ψ), based on the amino acid mis-incorporated in the study of Eyler et.al. (2019). Nucleotides in green indicate a mismatch in front of Ψ. Nucleotides in red indicate a mismatch next to Ψ. N: A, U, C or G base. Possibilities of codon-anticodon pairings with more than one mismatch are not represented.

## Data Availability

Not relevant.

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
