# Peer review of "The Importance of the Epi-Transcriptome in Translation Fidelity"

_ncrna, 2021, doi:10.3390/ncrna7030051_

Round 1

Reviewer 1 Report

This is an excellent and timely review on the role of RNA modification in protein synthesis, with specific focus on translational fidelity. Authors adequately describe role of RNA modifications found in three classes of RNAs, namely rRNAs, tRNAs and mRNAs. I only have one major and two minor suggestions to be included into the manuscript:

Major: I think that connection between cellular stress and RNA modifications in tRNAs, tRFs, mRNAs (localization to Stress Granules), and rRNAs is not described. Also, general overview of translation control mechanisms would be beneficial here (e.g., authors mention eIF2-alpha phosphorylation but do not describe this stress response pathways in sufficient details)

Minor:

1) I believe that authors should more critically discuss the obstacles to study RNA modifications, e.g. difficulties to detect and quantify RNA modifications in tRNAs (e.g. tRNA-seq technique) or specificity of antibodies.

2) Connection between tRNA modicifations and tRNA cleavage as well as role of tRFs in translational control (and RNA modifications on them) should be discussed

Reviewer 2 Report

The work by Valadon and Namy revises the current knowledge on RNA modifications and their role in translation fidelity. While the review is meant to focus on tRNA and rRNA modifications (page 3, first paragraph), I feel that the section on tRNA modifications could be improved by including some important aspects that significantly affect translation fidelity that are currently missing in the manuscript. I highlight below some of these aspects that I encourage the authors to consider to improve the manuscript.

  1. Page 3, last sentence of introduction: “There is a striking difference (…) via reader proteins”. The authors might want to consider softening this claim given that, at least for tRNA modifications, they can also act via “reader” proteins such as aaRS (e.g. tRNA modifications can serve as identity elements for cognate aaRS, and misacylation of tRNAs can lead to defects in translation fidelity).
  2. Related to comment 1. It would be helpful if the authors discuss tRNA modifications that are important for correct aminoacylation.
  3. Perhaps the authors might want to consider discussing the (few) cases of tRNA modifications away from the anticodon loop that seems to play a role in translation (see for example Hsin-Jung Chou et al Mol Cell 2017).
  4. The authors should discuss other important tRNA modifications within the anticodon loop that also affect translation fidelity that are currently missing from the manuscript such as complex modifications at position 37 (ms2t6A, yW, etc.) (see for example Tuorto and Lyko Open Biology 2016 for key references). Some of these modifications have also been associated to human diseases. A classic example is CDKAL1 deficiency leading to reduced ms2t6A modification in tRNA-Lys which affects translation of Lys codons in pre-proinsulin and prevents its maturation into insulin leading to type 2 diabetes (Fan-Yan Wei et al J Clin Invest 2011).
  5. Related to Comment 4. Many modifications at position 37 are important to prevent ribosomal frameshifting (which can also lead to detrimental translation fidelity). The authors could discuss the role of tRNA modifications in controlling ribosomal frameshifting. Not only modifications at position 37 affect this process, but also other modifications which are discussed in this manuscript (e.g. mcm5U).
  6. Mounting evidence now suggest that tRNA modifications within the anticodon loop are biologically interconnected (Han and Phizicky RNA 2018). It is possible then that the role of tRNA modifications in decoding is not only driven by a single modification but rather a network of modifications within the tRNA molecule. It would be interesting if the authors briefly discuss this concept and potential roles of cooperative tRNA modifications in translation (see for example William A. Cantara et al J Mol Biol 2012 or Clement T.Y. Chan et al Chem Res Toxicol 2015).
  7. The claim on page 8, last sentence of the first paragraph needs to be softened given the realization that ALKBH1 is a tRNA demethylase that regulate translation acting as a tRNA modification “eraser” (Fange Liu et al Cell 2016).
  8. Section 4.4 is a bit confusing. Only recently the enzymes responsible for m5C modification on mRNAs are being characterized. Perhaps it will be helpful if the authors include a brief introductory paragraph explaining the role of the NSUN family of proteins in m5C methylation of RNAs in general. Otherwise, for the non-specialized reader, it is hard to follow the discussion on NSUN2 (that modifies mRNAs but also tRNAs) followed by NSUN6 (that has usually been considered a modification enzyme for few tRNAs and only recently it has been shown to modify mRNAs).

Minor comments:

  1. Page 3, second paragraph. The authors define the “wobble position” as the third position of the codon. Technically, the wobble position is the first position of the tRNA anticodon (that can structurally “wobble” to pair with the third position of the mRNA codon). I would recommend the authors to rephrase the sentence as “Codon-anticodon pairing is known to be flexible at the third position of the codon, but it is clear that RNA modifications alter translation accuracy”.
  2. Figure 2. For convention, inosine modification is written with capital letter (I) while the lower case is reserved for isopentyl modifications (e.g. i6A). I recommend to change the “i” for “I” in Figure 2 and its Figure legend.
  3. All sections of the manuscript have a sentence or two on human diseases associated to the RNA modifications being discussed, except the section on mcm5U34. There are several mutations on genes encoding proteins from the human elongator complex that have been linked to neurological disorders that the authors might want to mention in this review.
  4. A recent article on translation by tRNAs with I34 has been published (Torres AG et al Nucleic Acids Res 2021) that the authors might want to consider for complementing the last paragraph of the section on inosine modifications.
  5. There is a formatting issue in section 2.4. The title is not aligned with the paragraph and the first paragraph is written in italics.

Reviewer 3 Report

In this review manuscript, the authors summarize the current knowledge of the link between RNA modifications and translation fidelity. The manuscript contains two main sections that are dedicated to modified nucleotides found in tRNAs and rRNAs respectively. However, another short section on mRNA modification is also presented. The manuscript is illustrated by 3 figures. In general, the topic addressed by this review is timely and of great interest for the general fields of RNA and translation. The text is generally illustrated properly by figures and refers to 137 references. Unfortunately, many of these references are sometimes too old and several specific points need to be carefully updated by the most recent references before acceptance.

For instance:

- the single reference for translation mechanisms is from 2004, this is definitely not up-to-date. There are numerous excellent reviews on translation mechanisms that were published recently. No reference is cited for RLI1/ABCE1, here again there is a recent review. No reference for IRES.

- most of the ground-breaking publications from the Sebastian Leidel lab including publications on U34 modification and its impact on translation fidelity are missing. In addition, a review very similar to that one is not cited (PMID: 31206634).

- recent references for modified nucleotides in human rRNA are missing, for example PMID: 29143818, PMID: 30202881

Other specific concerns:

- the nomenclature that is generally used for tRNA is: tRNAArg or tRNASec, in the manuscript the authors use Arg-tRNA and tRNAArg, please check and homogenize the nomenclature throughout the whole manuscript.

- In general the text requires careful proofreading to avoid ambiguous sentences or misleading points. For instance, in the sentence p3:

‘It begins with the binding of the 40S ribosomal subunit to the 5’ cap of the messenger RNA … ‘

This is not correct it’s a shortcut, the 40S does not bind to the cap, the eIF4F complex is bound to the cap and allows the subsequent recruitment of the 43S complex, which contains the 40S.

The sentence lacks precision and contains mistakes; it has to be remodelled accordingly.

Such erroneous sentences are found throughout the whole manuscript and require careful proofreading.

- The mains focuses of the manuscript are modified nucleotides in tRNAs and rRNAs, this does not appear clearly in the abstract. The same remark applies to the title that is misleading.

- The references list has to be carefully checked for typos and format mistakes.

- The first half of paragraph 2.4 is in italic.

Round 2

Reviewer 2 Report

The authors have successfully addressed all the comments raised by this Reviewer and the manuscript has improved considerably.

Minor comment: There is a reference not cited as a number in the main text (page 8, section i6A37, Blanchet 2017).

Reviewer 3 Report

The authors addressed all my concerns satisfyingly except for references concerning ABCE1, reference #24 is not appropriate and should be replaced by a recent review on ABCE1 instead (PMID: 33289941). After this last correction, I can now recommend this manuscript for publication in Non-coding RNA.